# Exploring Radiologists’ Burnout in the COVID-19 Era: A Narrative Review

**DOI:** 10.3390/ijerph20043350

**Published:** 2023-02-14

**Authors:** Michela Gabelloni, Lorenzo Faggioni, Roberta Fusco, Federica De Muzio, Ginevra Danti, Francesca Grassi, Roberta Grassi, Pierpaolo Palumbo, Federico Bruno, Alessandra Borgheresi, Alessandra Bruno, Orlando Catalano, Nicoletta Gandolfo, Andrea Giovagnoni, Vittorio Miele, Antonio Barile, Vincenza Granata

**Affiliations:** 1Nuclear Medicine Unit, Department of Translational Research, University of Pisa, 56126 Pisa, Italy; 2Academic Radiology, Department of Translational Research, University of Pisa, 56126 Pisa, Italy; 3Medical Oncology Division, Igea SpA, 80013 Naples, Italy; 4Department of Medicine and Health Sciences V. Tiberio, University of Molise, 86100 Campobasso, Italy; 5Department of Emergency Radiology, Careggi University Hospital, 50134 Florence, Italy; 6Italian Society of Medical and Interventional Radiology, SIRM Foundation, 20122 Milan, Italy; 7Department of Precision Medicine, Università degli Studi della Campania “Luigi Vanvitelli”, 80138 Naples, Italy; 8Area of Cardiovascular and Interventional Imaging, Abruzzo Health Unit 1, Department of Diagnostic Imaging, 67100 L’Aquila, Italy; 9Department of Radiology, University Hospital “Azienda Ospedaliera Universitaria delle Marche”, 60126 Ancona, Italy; 10Department of Clinical, Special and Dental Sciences, Università Politecnica delle Marche, 60126 Ancona, Italy; 11Department of Radiology, Istituto Diagnostico Varelli, 80126 Naples, Italy; 12Diagnostic Imaging Department, Villa Scassi Hospital-ASL 3, 16149 Genoa, Italy; 13Department of Biotechnological and Applied Clinical Sciences, University of L’Aquila, 67100 L’Aquila, Italy; 14Division of Radiology, Istituto Nazionale Tumori IRCCS Fondazione Pascale—IRCCS di Napoli, 80131 Naples, Italy

**Keywords:** burnout, stress, radiology, COVID-19

## Abstract

Since its beginning in March 2020, the COVID-19 pandemic has claimed an exceptionally high number of victims and brought significant disruption to the personal and professional lives of millions of people worldwide. Among medical specialists, radiologists have found themselves at the forefront of the crisis due to the pivotal role of imaging in the diagnostic and interventional management of COVID-19 pneumonia and its complications. Because of the disruptive changes related to the COVID-19 outbreak, a proportion of radiologists have faced burnout to several degrees, resulting in detrimental effects on their working activities and overall wellbeing. This paper aims to provide an overview of the literature exploring the issue of radiologists’ burnout in the COVID-19 era.

## 1. Introduction

Burnout is the experience of emotional exhaustion, depersonalization, and reduced personal accomplishment emerging as a prolonged response to chronic interpersonal stressors on the job [1]. It can affect the social and personal wellbeing of the professionals involved, resulting in psychological consequences (such as depression and a higher risk of suicide) that may have a negative impact on their quality of life, and on a larger scale, lead to reduced productivity and inflated healthcare costs [2,3,4]. 

The 2020 Medscape National Physician Burnout and Suicide Report revealed that 46% of radiologists are burned out, as compared with 42% of all physicians [5]. Triggers of stress and burnout for radiologists include heavier workload and complexity of cases, work–life imbalance, dysfunctional workplace dynamics, computer failure and consequently lack of control of their work, discontinuities in workflow (due, e.g., to ambient sounds from equipment or computers, noise related to background conversations, and working in an inpatient or emergency room setting), and social isolation [6,7,8,9,10,11,12,13,14].

On 11 March 2020, the World Health Organization declared the COVID-19 outbreak a pandemic. COVID-19 is a respiratory illness with a broad clinical spectrum, ranging from mild to moderate disease to severe disease and critical illness, which has not only proven to be a medical emergency but has also wreaked havoc on world societies and economies with lingering consequences in the foreseeable future [15,16,17].

Radiology units have been at the forefront of the emergency because of interstitial pneumonia (the most common presentation of COVID-19) and its potential complications, whose diagnosis and clinical evaluation rely heavily on radiological imaging, with special reference to multidetector computed tomography. Radiology departments have faced the reorganization of workload, having to guarantee the management of acute emergencies (while granting continuing assistance to cancer patients, fragile ones, or those with chronic conditions), and in parallel, the diagnostic and interventional management of COVID-19 patients. Furthermore, radiology units have been forced to create separate paths for COVID-19 and non-COVID-19 patients to avoid cross-contamination and to train all involved healthcare operators on the use of personal protection equipment (PPE).

The patients’ fear of becoming infected inside hospitals and the cancellation of most visits and non-urgent surgeries imposed by governments have brought a decline in outpatient procedures, along with substantial financial cuts and job losses in private facilities [18,19,20,21,22,23,24,25,26,27,28,29,30,31,32,33,34,35,36,37,38,39,40,41,42,43,44,45,46,47,48,49]. Among physicians, and especially radiologists, the COVID-19 pandemic has exacerbated burnout stressors and brought to light new triggers, such as fear of contracting the COVID-19 infection, anxiety about the future of their families or communities, scarcity of PPE, longer shifts, and increased social isolation [50,51,52,53,54,55,56,57,58,59,60]. In a national survey conducted by Coppola et al. on the impact of the COVID-19 crisis on the professional and personal wellbeing of radiologists, more than 50% of respondents complained about having higher emotional stress at work along with moderate or severe symptoms (including sleep disturbances, feeling like living in slow motion, and having negative thoughts), with more specific criticalities in the country areas more severely struck by the COVID-19 outbreak [61].

Our aim is to review the current literature dealing with radiologists’ burnout during the COVID-19 pandemic.

## 2. Materials and Methods

A systematic literature search was carried out to analyze radiologists’ burnout in terms of causes, incidence, and possible remedies in the COVID-19 era. We searched the PubMed (https://pubmed.ncbi.nlm.nih.gov (accessed on 8 December 2022)) and Scopus (https://www.scopus.com (accessed on 8 December 2022)) databases by using a combination of the following search terms: ((“stress in radiologist” OR “burnout in radiologist” OR “demotivation in radiologist”) AND (“stress in radiology” OR “burnout in radiology” OR “demotivation in radiology”)). The search period ranged from March 2020 to December 2022, and the search was performed on 8 December 2022. Only articles published in English language were included. 

To improve the quality of our inclusion criteria, our analysis was conducted following the Preferred Reporting Items for Systematic Reviews and Meta-Analyses (PRISMA) guidelines [62]. 

## 3. Results

Two reviewers (M.G., L.F.) manually screened all articles retrieved (*n* = 182) by title and abstract, removed all duplicate records (*n* = 5) and if potentially eligible, their full text was reviewed. Exclusion criteria were as follows: (1) articles reporting data collection carried out before 2020 (*n* = 10); (2) review, editorial, and commentary articles (*n* = 15); (3) no access to the article’s full text (*n* = 8). 

A total of 122 studies were excluded due to having irrelevant titles and abstracts, and the remaining ones underwent a more thorough analysis. Finally, 22 articles were singled out and divided into the following categories: (1) stress emerged during the COVID-19 outbreak (*n* = 10), (2) stress unrelated to the COVID-19 outbreak (*n* = 2), and (3) strategies to enact as possible remedies to burnout (*n* = 10) (Figure 1). 

The articles analyzed report the findings from surveys and the remaining articles are summarized in Table 1 and Table 2, respectively.

## 4. Discussion

To recognize and address burnout, physicians should be aware of its manifestations and demographic variability. A contribution to burnout may derive from the so-called imposter phenomenon, which has a high rate of prevalence in medical professionals and refers to a feeling of inadequacy and self-doubt due to the inability to internalize and accept success and skill. Deshmukh et al. [82] investigated the occurrence of burnout in radiologists and reported a statistically significant correlation (*p* = 0.024) between burnout and the imposter phenomenon, with 83% of surveyed clinical radiology faculty reporting feelings of imposter phenomenon during their career. Possible explanations include the rapid development of technologies in the field, the quick pace of daily workflow, and the lack of positive reinforcement via patient interactions.

During the COVID-19 pandemic, work-related stress ramped up because of mandatory changes [48,83,84,85,86,87,88,89,90,91,92,93,94]. Several surveys [61,64,65,67,68,70,71,95] conducted during the COVID-19 outbreak in different countries have revealed a sensible impact of it on radiologists’ life in terms of anxiety, fear of spreading the infection, difficulty sourcing PPE, shifts in workload due to an increase in emergency procedures, and feeling that the crisis had affected their family relationships and damaged their own and their colleagues’ professional activity. In a survey conducted among North American pediatric radiologists by Ayyala et al. [63], 69% of respondents endorsed feeling more isolated from a lack of regular interaction with colleagues due to an increase in remote working. A balance between work and personal life was a critical issue (with 53% of respondents indicating that it is challenging to work remotely while overseeing home schooling for children), resulting in conflicting situations between at-work and outside-work duties.

Furthermore, concerns were expressed by trainees and educators regarding the effectiveness of education programs limited by social distancing, lack of regular clinical activities, difficulty changing residents’ rotations or postponement of core examinations [61,64]. However, Huang et al. [66] showed that compared to nonradiologists, radiology staff reported lower mean scores of anxiety (4.0 vs. 4.9, *p* = 0.035) and burnout (1.9 vs. 2.1, *p* = 0.002). Moreover, among radiology staff, anxiety and burnout were associated with longer than usual working hours (*p* < 0.01), and higher job dedication was associated with lower anxiety (*p* < 0.01). Such findings could be related to sufficient reserves of PPE, rapid testing, and ready access to up-to-date information, but also to stronger job dedication and teamwork.

Recent papers have revealed that women are more susceptible to burnout than men, particularly in relation to the COVID-19 outbreak [79,80,96], also due to being subjected to more stressors related to familial duties. Huang et al. [67] found that the resilience of female medical staff in radiology departments during the COVID-19 outbreak was significantly lower than that of male medical staff. In a study by Milch et al. involving breast radiologists [69], psychological distress was reported to be highest among females, younger professionals, and those with greater pandemic-specific childcare needs and financial losses, threatening to exacerbate existing gender disparities in radiology. More generally, Ayyala et al. [63] reported that compared to men, female radiologists suffered from higher work-related stress and anxiety (*p* = 0.02), greater feelings of guilt from radiology technologists and nurses being more exposed to COVID-19 (*p* = 0.02), and higher levels of stress providing for dependents (*p* = 0.04). Conversely, in the Italian nationwide survey by Coppola et al., women from areas more severely hit by the COVID-19 crisis were less susceptible to feeling like living in slow motion than those from other areas, possibly related to a greater sense of control in workplaces more directly affected by the crisis, which could foster engagement and help avoid burnout [61,97,98].

The COVID-19 outbreak has unraveled the advantages of remote working, but also its disadvantages in terms of isolation, especially for pediatric radiologists, whose professional activity entails more patient interaction than other radiology subspecialties [63]. In the last years, radiologists have assumed a more active role in patient care by taking part in team decision-making, thus clarifying the diagnostic strategy and refining the therapeutic decisions of clinician members [99,100]. In keeping with Brown et al. [74], the distancing requirements caused by the COVID-19 pandemic have emphasized the value of interdisciplinary collaboration, suggesting that cultivating relationships with patients and colleagues could be key to fighting off burnout through a “wholeness” (rather than personal “wellness”) approach.

The causes of burnout are probably multifactorial, and therefore it is unlikely that there can be a single remedy. General strategies to combat these issues may include education on existing evidence-based practice guidelines, and utilization of computer-based clinical decision support systems to perform appropriate radiological procedures, avoid redundant imaging and streamline imaging protocols. Furthermore, excessive workload and understaffing in healthcare environments can add to work-related stressors and possibly lead to burnout, whereas maintaining an adequate supply of physicians is vital to a well-functioning healthcare system. Fostering innovation in radiology departments (such as the widespread adoption of radiological structured reporting, the use of validated AI systems to assist radiologists’ activity, the implementation of optimal ergonomics, and the constructive use of social media) can also play a role in preventing burnout by optimizing workflow and improving overall work quality [77,78,81,101,102,103,104,105,106,107,108,109,110,111,112,113,114,115,116,117,118,119,120,121,122,123,124,125,126,127,128,129,130,131,132,133,134,135,136].

Another approach to burnout could be improving the mental status of radiologists by building resilience and practicing mindfulness [72]. Mindfulness has its roots in Buddhism, in which practitioners pay attention to each moment with openness and acceptance without judgment and turn their minds away from the stress of daily life [76,137]. Buch et al. [75] illustrated the results of the implementation of a faculty wellness program in an academic radiology department before the COVID-19 pandemic and evaluated the change in such initiatives following the pandemic. In the pre-COVID-19 era, their program focused on improving efficiency, limiting disruptions, improving reading rooms and ergonomics, creating a more efficient workflow, and opportunities for social interaction and team building. After the onset of the COVID-19 pandemic, radiologists requested more initiatives focused on wellbeing, emotional health, mindfulness, and personal access to mental healthcare. From this viewpoint, according to Fishman et al. [76], coaching is an emerging and promising approach to address the burnout epidemic affecting radiologists. Coaching can be defined as “partnering with clients in a thought-provoking and creative process that inspires them to maximize their personal and professional potential”. Professional coaches help clients, who already possess the skills needed to overcome the challenge they face, to navigate their professional and personal journeys through better steering of talents, resources, and desires [138,139]. 

Belfi et al. [73] reported a positive impact of storytelling on radiologists’ perceived sense of wellbeing, shifting their sense of empathy and connectedness to others during the COVID. Nineteen Association of University Radiologists (AUR) members were selected for participation in Storytelling Geek Week, a virtual workshop that took place over five days in November 2020. Participants’ current state of wellbeing was significantly increased between the pre- and post-course surveys (*p* = 0.017), and respondents reported shifts in their sense of empathy and connectedness to others, suggesting that storytelling may help mitigate burnout and build community during challenging times.

## 5. Conclusions

Burnout is a complex phenomenon that affects radiologists as members of the medical community and has seen a surge during the COVID-19 outbreak due to its dramatic impact on public health, extending beyond working environments to the family and personal lives of healthcare professionals. Radiologists have worked in the frontline of healthcare systems during the COVID-19 pandemic, experiencing a disruption in their work activities, especially during the first wave of the crisis. This has resulted in an increased vulnerability to burnout, and in several cases has exacerbated criticalities in radiology workplaces hailing back to the pre-COVID-19 era, leading to some manifestations that are specific to the radiologist’s profession. In line with Kruskal et al. [139], organizations and leaders will need to reprioritize efforts to build high-functioning cohesive teams, implement peer-support practices, and support posttraumatic growth, while optimizing meaning in work. A multipronged approach that involves optimizing resources, improving workload efficiency, adapting technology-based solutions, and promoting wellbeing (both as individuals and among colleagues working in the same environment with the goal of promoting patients’ health) could be key to preventing the risk of burnout while improving job quality, efficiency, and overall satisfaction.

## Figures and Tables

**Figure 1 ijerph-20-03350-f001:**
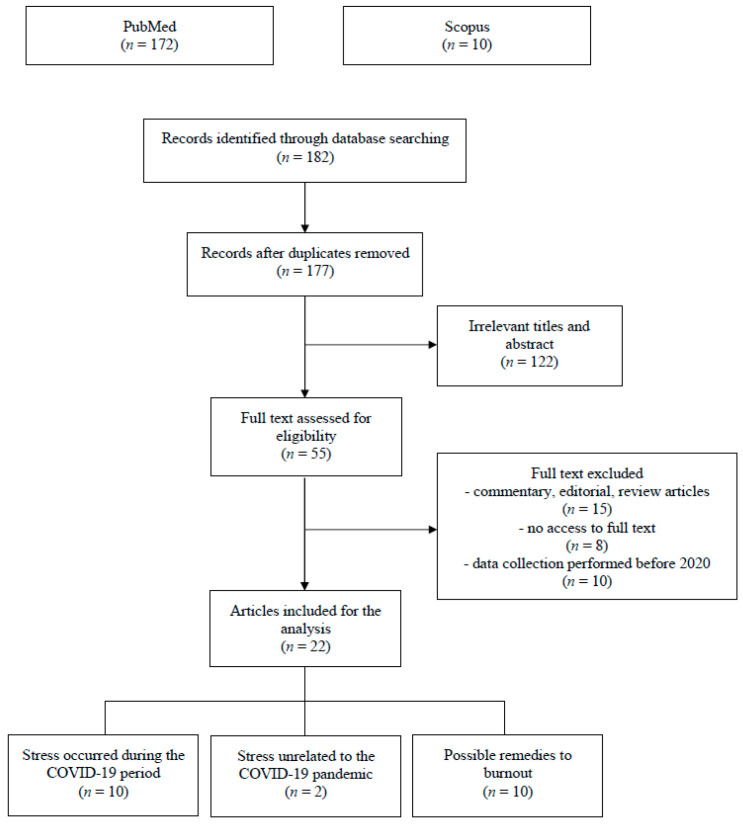
PRISMA flow chart illustrating the inclusion and exclusion criteria for the article search.

**Table 1 ijerph-20-03350-t001:** Articles analyzed based on surveys. CI_95%_ = 95% confidence interval. OR = odds ratio. RR = relative risk. SD = standard deviation. *authors’ personal communication.

Authors [Reference Number]	Year	Country	Qualification	Age (Years)	Professional Experience (Years)	Gender Distribution	Survey Duration	Main Findings
Ayyala et al. [63]	2021	North America	Physician members of the Society for Pediatric Radiology	Median 48 (range 33–70)	Median 14 (range 1–45)	Female 53%, male 46%	From 27 April 2020 to 22 May 2020	Response rate 21% 69% of respondents felt more isolated from a lack of regular interaction with colleagues 53% of respondents indicated that it is challenging to work remotely while overseeing home schooling for children In comparison to men, women reported overall higher work-related stress and anxiety (*p* = 0.02), higher feelings of guilt from radiology staff (i.e., technologists and nurses) being more exposed to COVID-19 (*p* = 0.02) and higher levels of stress providing for dependents (*p* = 0.04)
Coppola et al. [61]	2021	Italy	Members of the Italian Society of Medical and Interventional Radiology (SIRM)	Under 35 (18.7%) 36–65 (68.2%) Over 65 (13%) (“hot regions”) Under 35 (16.3%) 36–65 (70.8%) Over 65 (12.8%) (other regions)	Resident (10.7%) Post-doc fellow, PhD student or outpatient specialist (1.2%) Research fellow (0.4%) Associate or full professor (1.3%) Staff radiologist (52.2%) Medical director (8.5%) Private consultant (25.7%) (“hot regions”) Resident (9.8%) Post-doc fellow, PhD student or outpatient specialist (1.4%) Research fellow (1.2%) Associate or full professor (1.4%) Staff radiologist (52.6%) Medical director (10.8%) Private consultant (21.8%) (other regions)	Female 50.7%, male 49.3% (“hot regions”) Female 51.3%, male 48.7% (other regions)	From 4 May 2020 to 11 May 2020*	Response rate 20.1% More than 60% of respondents estimated a workload reduction greater than 50%, with a higher prevalence among private workers in hot regions (72.7% vs. 66.5% elsewhere, *p* = 0.1010) More than 40% were moderately or severely worried that their professional activity could be damaged, and most residents believed that their training had been affected More than 50% of respondents had increased emotional stress at work, including moderate or severe symptoms due to sleep disturbances, feeling like living in slow motion and having negative thoughts, those latter being more likely in single-living respondents from hot regions [log OR 0.7108 (CI_95%_ 0.3445 ÷ 1.0770), *p* = 0.0001]
Demirjian et al. [64]	2020	USA	Radiologists	Mean 45 (SD 11)	Radiologist/attending radiologists (80%) In training (residents) (16%) In training (fellows) (4%) Other (0.4%)	Female 47% male 53% nonbinary 0.2% prefer not to say 0.7%	From 3 April 2020 for 8 days	61% of respondents rated their level of anxiety with regard to COVID-19 as 7 out of 10 or greater, and higher scores were positively correlated the standardized number of COVID-19 cases in a respondent’s state (RR = 1.11, CI_95%_ 1.02–1.21, *p* = 0.01) Citing the stressor of “personal health” was a strong predictor of higher anxiety scores (RR 1.23, CI_95%_ 1.13–1.34, *p* < 0.01) Participants who reported needing no coping methods were more likely to self-report lower anxiety scores (RR 0.4, CI_95%_ 0.3–0.53, *p* < 0.01)
Elshami et al. [65]	2021	Middle East, North Africa, India	Radiographers (92.5%) radiologists (2.3%) advanced practitioners (1.4%) radiology residents (1.0%) radiology assistants (0.9%) radiology nurses (0.9%) others (1.0%)	18–29 (49.7%) 30–39 (27.9%) 40–49 (17.8%) 50–59 (4.2%) 60+ (0.3%)	Less than 5 (38.1%) 6–10 (24.6%) 11–15 (14.0%) 16–20 (11.0%) 20+ (12.4%)	Female 49.1% male 50.9%	From 22 May 2020 to 2 June 2020	Respondents reported experiences of work-related stress (42.9%), high COVID-19 fear score (83.3%) and anxiety (10%) during the COVID-19 pandemic 79.5% of respondents strongly agreed or agreed that PPE was adequately available at work during the pandemic It is important to provide training and regular mental health support and evaluations for healthcare professionals, including radiology workers, during similar future pandemics
Huang HL et al. [66]	2021	Singapore	Doctors (21.7%) nurses (2.2%) allied health professionals (58.9%) others (17.2%)	Mean 38.6 (SD 12.4)	N/A	Female 62.8% male 37.2%	From 12 March 2020 to 20 July 2020	During the COVID-19 pandemic, a proportion of radiological staff reported significant burnout and anxiety, although less compared to the larger hospital cohort (17.8% vs. 23.9%, *p* = 0.068) and 6.7% vs. 13.2%, *p* = 0.013) Measures to prevent longer than usual work hours and increase feelings of enthusiasm and pride in one’s job may further reduce the prevalence of anxiety problems and burnout in radiology departments
Huang L et al. [67]	2020	China	Doctors (38.0%) technicians (41.7%) nurses (20.3%)	Median 33 (range 28–43)	Median 10 (range 5–21)	Female 52.0% male 48.0%	From 7 February 2020 to 9 February 2020	Effective response rate 97.8%. The resilience level of the medical staff in the radiology departments during the COVID-19 pandemic was generally low, particularly regarding toughness More attention should be paid to resilience influence factors such as high perceived stress, female gender, lack of understanding of COVID-19 and protective measures, and lack of protective materials, and targeted interventions should be undertaken to improve the resilience level of the medical staff in the radiology departments during the outbreak of COVID-19
Huang L et al. [68]	2020	China	Doctors, technicians, nurses	Median 33 (range 28–43)	Median 10 (range 5–21)	Female 52.0% male 48.0%	From 7 February 2020 to 9 February 2020	Medical staff in radiology departments faces a higher risk of infection and a heavier workload during the COVID-19 pandemic Risk factors for perceived stress were female gender, existing anxiety, and fears of being infected at work, an uncontrollable outbreak, and not being able to pay rent or mortgage
Milch et al. [69]	2021	USA (97%), Canada (1%), other countries (1%)	Members of the SBI (Society of Breast Imaging) and the National Consortium of Breast Centers (NCBC)	31–40 (19%) 41–50 (26%) 51–60 (34%) 61–70 (20%) 71+ (2%)	<5 (11%) 5–10 (18%) 11–20 (19%) 20+ (52%)	Female 79% male 21%	From 29 June 2020 to 18 September 2020	Overall response rate 18%. Anxiety was reported by 68% of respondents, followed by sadness (41%), sleep problems (36%), anger (25%), and depression (23%) A higher psychological distress score correlated with female gender (OR 1.9, *p* = 0.001), younger age (OR 0.8 per SD; *p* = 0.005), and a higher financial loss score (OR 1.4, *p* < 0.0001) Participants whose practices had not initiated wellness efforts specific to COVID-19 (54%) had higher psychological distress scores (OR 1.4, *p* = 0.03) Of those with children at home, 38% reported increased childcare needs, higher in women than men (40% vs. 29%, *p* < 0.001). A total of 37% reported that childcare needs had adversely affected their jobs, which correlated with higher psychological distress scores (OR 2.2–3.3, *p* < 0.05)
Oprisan et al. [70]	2021	Spain		Mean 40.39 (SD 10.8)	0–4 (20.3%) 5–9 (25%) 10–19 (29.1%) 20–29 (18.9%) > 30 (6.7%)	Female 58.7% male 41.3%	From April to August 2020	The prevalence of burnout syndrome increased during the COVID-19 pandemic (49.3% vs. 33.6%, *p* = 0.002) No risk factors or protective factors that were valid both before and after the pandemic were identified No correlations were identified between sociodemographic or work-related characteristics and burnout syndrome
Woerner et al. [71]	2021	USA (77.5%) India (9.7%) UK (5.9%) Saudi Arabia (2%) Spain (1%) Chile (1%) other (3%)	Interventional radiologists (Society of Interventional Radiology, SIR)	N/A	78.9% interventional radiologists 21% interventional radiologists in training	Female 15.9% male 83.9%	From 28 November 2020 to 23 December 2020	The COVID-19 pandemic induced practice alterations and high rates of self-reported anxiety in interventional radiology Female gender, increased call coverage, and lack of adequate or timely departmental adjustments were associated with increased anxiety levels

**Table 2 ijerph-20-03350-t002:** Articles analyzed not based on surveys.

Authors [Reference Number]	Year	Objective
Aggarwal et al. [72]	2021	To discuss several strategies for mitigating high volumes, including abbreviated MRI protocols, 24/7 radiologist coverage, reading room assistants, and other strategies to tackle radiologist burnout
Belfi et al. [73]	2022	To investigate the process of storytelling as a self-care practice for radiologists
Brown et al. [74]	2021	To introduce wholeness, rather than wellness, to address the symptoms of burnout To focus radiologist as member of health care teams, rather than an individual, as a solution to the problem
Buch et al. [75]	2021	To implement a wellness program in an academic radiology department to prevent burnout and to assess initial outcomes, with special focus on the challenges related to the COVID-19 pandemic
Fishman et al. [76]	2021	To evaluate coaching as a method to combat burnout
Glover et al. [77]	2022	To evaluate how optimizing radiologist’s work environment to improve overall quality of life and wellness
McGrath et al. [78]	2022	To enhance radiologists’ productivity and efficiency through optimization of institutional infrastructure, reading room and workstation, user-level interactions with personal devices, and advances in artificial intelligence
Oliveira et al. [79]	2022	To evaluate key techniques at the individual, peer, and institutional levels to offer a multifaceted approach to ameliorating radiologist burnout
Parikh et al. [80]	2021	To describe common misperceptions that may contribute to radiology practice leaders not addressing burnout and to explore practical skills that leaders should develop to effectively address burnout
Patel et al. [81]	2021	To abbreviate musculoskeletal MRI protocols
Parikh et al. [80]	2021	To evaluate the major stressors affecting breast radiologists
Deshmukh et al. [82]	2022	To investigate imposter phenomenon, and assess correlation with burnout, in radiologists. To pilot an intervention aimed at addressing imposter phenomenon through improvisational theater techniques

## Data Availability

Data can be made publicly available upon reasonable request.

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
