# Peer review of "Exploring Radiologists’ Burnout in the COVID-19 Era: A Narrative Review"

_ijerph, 2023, doi:10.3390/ijerph20043350_

Round 1
Reviewer 1 Report
The paper presented for review, in accordance with the title, is to discuss the results of research on the occurrence of burnout syndrome in radiologists during the Covid-19 pandemic, although it actually discusses the general occurrence of this phenomenon in a rather disordered manner, and only marginally considers the impact of the covid-10 pandemic, indicating mostly to factors not specific to the profession of a radiologist. A total of 22 publications were analyzed. The results chapter lacks a detailed discussion of the articles, e.g. in terms of the region in which the study was conducted, the size of the study groups, the age of the respondents, place of work, seniority, as these factors may affect the occurrence of burnout syndrome. A review article should organize and systematize the knowledge and results obtained by individual authors, but this work lacks this. Discussing requires a comprehensive organization of information obtained from a systematic review, not a discussion of subsequent articles.
Author Response
Thank you for your suggestions. To present the relevant study findings in a clearer and more systematic manner, we added two tables, of which one (Tab. 1) summarizing the key findings of each survey article along with the country of origin and characteristics of the study groups (including qualification, professional experience and gender distribution), and the other one (Tab. 2) summarizing the key findings of the remaining articles. In addition, we streamlined the Discussion section with a stronger focus on the factors related to COVID19-related burnout that are more specific to radiologists.
Reviewer 2 Report
Dear Authors,
The authors have written a well structured narrative review . They have explained and elaborated the radiologists’ burnout in the Covid-19 era. However, I feel they should have included one table summarizing the main findings of the included studies .
Regards
Author Response
Thank you very much for your appreciation. Following your advice and another reviewer's suggestions, we added two tables, of which one (Tab. 1) summarizing the key findings of each survey article along with the country of origin and characteristics of the study groups (including qualification, professional experience and gender distribution of the survey respondents), and the other one (Tab. 2) summarizing the key findings of the remaining articles.
Reviewer 3 Report
I would like to thank the authors for this paper. I found it to be an interesting read because of the fact it addresses radiology specialty. I totally agree with statements dealing with stress & burnout at radiological environment regarding the outbreak of COVID-19 pandemic.
Authors have conducted the analysis following the Preferred Reporting Items for Systematic Reviews and Meta-Analyses (PRISMA) guidelines. They have clearly emphasized inclusion & exclusion criteria for the review. Beginning of the manuscript starts with a transparent introduction. Researchers gave us a very profound burnout description. Authors have prepared a legible presentation of their results and comprehensive discussion. Authors have prepared widespread reference section (133 positions). Majority of them are dated for 2020-2022 which is a real strength of up-to-date reviews.
I must admit that the review is neither innovative nor a novel, but it highlights very important issues we must bear in mind nowadays. Our mental health is at stake. It is called a silent epidemic of 21st century. Authors have stated that radiologists have worked in the frontline of healthcare systems at the peak of the COVID-19 pandemic, experiencing a disruption in their work activities that has resulted in an increased vulnerability to burnout.
Authors made a point of a multifaceted approach involving optimizing resources, improving workload efficiency, adapting technology-based solutions, and promoting wellbeing to be a key to prevent the risk of burnout while improving job quality, efficiency, and overall satisfaction.
Author Response
Thank you very much for your appreciation.
Reviewer 4 Report
The article “Exploring radiologists’ burnout in the Covid-19 era: a narrative review”, focuses on literature review and aims to reveal the situations of radiologists in the covid-19 period, related with burnout. The articles, object of literature review, intend to reveal the levels of stress emerged during the COVID-19 outbreak; stress unrelated to the COVID-19 outbreak and strategies to enact as possible remedies to burnout. Part of the principle, that radiologists suffer from burnout more than other health professionals.
Some points could be improved in the article:
Perhaps it is important to raise a starting question that guides the purpose of the article and that you draw the conclusions that it leads to. What is the specificity of burnout for radiologists? Is this burnout aggravated by Covid? What are its characteristics?
This type of question could elevate the article to another level of analysis that is not merely descriptive.
Na Line 144 – refute the idea that the review of the bibliography was carried out to "assess". Generally, a review of the literature is carried out to explore or analyze some issues.
The problematization and the methodology is sufficiently explained, but it lacked a point where the main characteristics of the two documents under review were presented.
On line 138, following the methodology, it would be recommended that we present the results of the bibliographic review: indicating authors, country of origin, data and focus of the article (for example, in the discussion it is revealed that there are specificities for radiologists: one of the mother or other children etc). The article would benefit from clarification of these data before discussing results.
The discussion of results could be shorter, focused specifically on responding to the objectives of the article. Perhaps some of the content of the discussion could be used to unequivocally demonstrate the results of the review carried out.
Saying this, the conclusion needs to be focused, do not be afraid and answer, explicitly, the starting questions and the objectives stated in the article.
I wish good luck and good work for the authors.
Author Response
Thank you for your insightful comments. As also suggested by other reviewers, we added two tables, of which one (Tab. 1) summarizing the key findings of each survey article along with the country of origin and characteristics of the study groups (including qualification, professional experience and gender distribution of the survey respondents), and the other one (Tab. 2) summarizing the key findings of the remaining articles. This has allowed us to shorten and further focus the Discussion section on the COVID19-related factors of burnout that are more specific to radiologists. With regard to the latter point, we also rephrased the Conclusions in a more focused manner.
Finally, we modified the statement at line 144 by replacing the term "assess" with the more neutral "analyze".
Round 2
Reviewer 4 Report
The revisions made by the authors greatly improve the article and therefore I recommend its publication.